# Focusing on Adenosine Receptors as a Potential Targeted Therapy in Human Diseases

**DOI:** 10.3390/cells9030785

**Published:** 2020-03-24

**Authors:** Wiwin Is Effendi, Tatsuya Nagano, Kazuyuki Kobayashi, Yoshihiro Nishimura

**Affiliations:** 1Division of Respiratory Medicine, Department of Internal Medicine, Kobe University Graduate School of Medicine, 7-5-1 Kusunoki-cho, Chuo-ku, Kobe, 650-0017, Japan; wisepulmo@gmail.com (W.I.E.); kkoba@med.kobe-u.ac.jp (K.K.); nishiy@med.kobe-u.ac.jp (Y.N.); 2Department of Pulmonology and Respiratory Medicine, Medical Faculty of Airlangga University, Surabaya 60131, Indonesia

**Keywords:** adenosine, adenosine receptors, G protein-coupled receptors, agonists, antagonists, allosteric molecules

## Abstract

Adenosine is involved in a range of physiological and pathological effects through membrane-bound receptors linked to G proteins. There are four subtypes of adenosine receptors, described as A_1_AR, A_2A_AR, A_2B_AR, and A_3_AR, which are the center of cAMP signal pathway-based drug development. Several types of agonists, partial agonists or antagonists, and allosteric substances have been synthesized from these receptors as new therapeutic drug candidates. Research efforts surrounding A_1_AR and A_2A_AR are perhaps the most enticing because of their concentration and affinity; however, as a consequence of distressing conditions, both A_2B_AR and A_3_AR levels might accumulate. This review focuses on the biological features of each adenosine receptor as the basis of ligand production and describes clinical studies of adenosine receptor-associated pharmaceuticals in human diseases.

## 1. Introduction

Adenosine is a nucleoside molecule that elicits various physiological responses in tissues and organs. The pioneering identification of adenosine occurred when Drury and Szent-Gyorgyi successfully extracted an adenylic acid substance from the mammalian heart and other tissues influencing cardiac rhythm [1]. Subsequently, adenosine analogs resulted in coronary dilatation and blood flow elevation in some studies [2]. Interestingly, a small amount of caffeine diminished the effect of adenosine on the contraction of atrial muscle [3]. In fact, Sattin and Rall proposed that adenosine required a particular molecule in the cell membrane to exert its effects [4]. All these studies considered the role of adenosine-related specific receptors.

The classical autonomic neurotransmitters released from peripheral nerves were once recognized as only noradrenaline (NA) and acetylcholine (Ach). The concept of noncholinergic and nonadrenergic transmissions was introduced after 5’-adenosine triphosphate (ATP) was recognized as a purinergic neurotransmitter [5]. Next, Burnstock designed two main types of purinergic receptors, i.e., P_1_ and P_2_, which are based on agonistic and antagonistic functions [6,7]. The affinity for adenosine of P_1_ was stronger than that of P_2_ [8,9]; therefore, receptors for adenosine were classified as P_1_, while ATP and 5´-adenosine diphosphate (ADP) were more suitable as natural ligands for P2 [10]. Based on the latest nomenclature of the International Union of Pharmacology Committee on Receptor Nomenclature and Drug Classification (NC-IUPHAR), the receptor for adenosine is named adenosine receptor (AR), which can be subdivided into four types: A_1_, A_2A_, A_2B_, and A_3_ [11]. These ARs are activated by endogenous and exogenous adenosine or its analogs [12].

## 2. Adenosine: Production, Transport, and Metabolism

Endogenous adenosine, a natural purine nucleoside consisting of the nucleobase adenine reacted with a sugar ribose by a glycosidic linkage [13], is a normal cellular element and is continuously produced, mainly intracellularly and extracellularly [11]. Adenosine is formed via dephosphorylation of its main source, nucleotide 5´-adenosine monophosphate (AMP), via both cytosolic 5´-nucleotidase (cN)-I and the inosine monophosphate (IMP)/guanosine monophosphate (GMP)-selective cN-II [14,15]. In addition, cN-I catalyzes AMP to adenosine, while cN-II plays a dominant role in the production of inosine and guanosine from IMP and GMP, respectively [16]. Intracellular adenosine is also generated by hydrolysis of *S*-adenosyl-homocysteine through the enzyme S-adenosyl-L-homocysteinase hydrolase [17]. Adenosine production from AMP is relatively faster than hydrolysis of *S*-adenosyl-homocysteine [18]. Adenosine may be found throughout endogenous purine synthesis [19]. If there is a mismatch between the production and use of ATP, for instance, in cases of hypoxia and ischemia, adenosine as well as other purine metabolites accumulate [20].

Intracellular adenosine can be released across the plasma membrane via bidirectional, concentrative nucleoside transporters (CNTs; sodium-dependent) and equilibrative nucleoside transporters (ENTs; sodium-independent) [21]. Based upon concentration gradients, ENTs are passive bidirectional transporters that transport adenosine across the plasma membrane, while CNTs are active Na^+^-dependent transporters [22]. ENTs are responsible for transporting adenosine in and out of the cell and are distributed in mammalian tissues [23].

Under normal conditions, the concentration of adenosine outside the cell is relatively low [24]. Three steps are necessary to produce extracellular nucleosides. First, the main source of extracellular adenosine is particularly generated from intracellular nucleotides, such as ATP, AMP, and ADP, which are released during stress, hypoxia, inflammation, or injury [19]. Intracellular ATP is an essential fuel to drive energy-requiring processes, such as active transport, cell motility, and biosynthesis [25]. The very abundant intracellular nucleotide ATP [26] will be released through exocytosis from vesicles and membrane transport proteins [27]. Potential candidates for particular transporter channels include cystic fibrosis transmembrane conductance regulators, multiple drug resistance channels, connexin hemichannels, maxi-ion channels, stretch-activated channels, and voltage-dependent channels [23].

Moreover, adenosine is produced by inflammatory cells, including mast cells [28], leucocytes [29], neutrophils [30], and eosinophils [31]. In accordance with the concept of retaliatory metabolites, increases in the level of extracellular adenosine regulate anabolic and catabolic hormones during stress [32]. Adenosine promotes the healing process after inflammation-induced injury [33]. Remarkably, extracellular ATP and adenosine are crucial alarms for cell alertness.

Extracellular nucleotides then undergo hydrolysis to remove the phosphate groups. Nucleotide and nucleoside degradation are determined by surface-located enzymes [34]. The ectonucleotidase cluster of differentiation (CD) 39, a lymphoid cell activation antigen bearing a resemblance to ATP diphosphohydrolase (ATPdase), catalyzes the dephosphorylation of ATP and ADP to form AMP [35,36]. Other ectonucleotidases and ecto-ATPases tend to catalyze ATP more than ADP [37]. CD39 is greatly expressed by the endothelium but not in resting T and B lymphocytes, natural killer (NK) cells, neutrophils, macrophages, and monocytes [38,39]. The transformation of ATP to AMP by CD39 can be reversed by the actions of the extracellular diphosphate kinases NDP kinase and adenylate kinase [40].

The final step in the production of extracellular adenosine is the dephosphorylation of AMP. Each purine and pyrimidine nucleoside monophosphate, especially AMP, is hydrolyzed by a glycosyl phosphatidyl inositol (GPI)-anchored enzyme, ecto-5´-nucleotidase [37]. Ecto-5´-nucleotidase is also recognized as CD73, which is found on both T and B lymphocytes [41].

The conversion of AMP to adenosine by CD73 is reversible only following intracellular transport of adenosine, after which it can be changed to AMP by adenosine kinase [40]. Moreover, adenosine can also be metabolized to inosine by adenosine deaminase (ADA) [42]. Overall, adenosine extracellular production depends on the balance between (i) its release from cells, (ii) its reuptake by bidirectional adenosine transport processes, and (iii) its conversion by ectonucleotidases on the cell surface [40].

## 3. Adenosine Receptors

Adenosine carries out different biological effects determined by each AR on the membrane surface of specific cells or tissues (Figure 1) [43]. Initially, ARs were grouped into A_1_ and A_2_ receptors based on their effects in inhibiting and stimulating cyclic AMP (cAMP) in the brain [44]. Currently, the four receptor subtypes (A_1_, A_2A_, A_2B_, and A_3_ receptors) have been purified and successfully cloned from mammalian and nonmammalian species, particularly rats, mice, and humans [11,43]. Based upon sequence similarity and G protein-coupling specificity, A_1_ and A_3_ receptor share 49% sequence identity and preferentially couple to Gα_i/o_ in the inhibition of adenylate cyclase (AC). In contrast, A_2A_ and A_2B_ receptors, which are almost 59% identical and prefer Gα_s_, are able to stimulate AC [45]. A_1_ and A_2A_ receptors possess high affinity, while A_2B_ and A_3_ receptors show relatively lower affinity.

ARs are members of heteromeric guanine nucleotide-binding protein (G protein)-coupled receptor (GPCR) family A [46]. All ARs have a seven-pass transmembrane α-helical structure with an extracellular amino terminus and an intracellular carboxyl terminus, and the *N*-terminal domain has *N*-glycosylation sites that influence trafficking of the receptor to the plasma membrane [22]. GPCRs, the largest class of cell surface receptors, are activated by various kinds of ligands, including hormones, neurotransmitters, ions, odorants, and photons of light, and couple to a wide range of signaling molecules and effector systems [47].

Receptors will interact with endogenous adenosine agonists (orthosteric ligands) or with molecules (allosteric ligands) that are located far from the orthosteric site [48]. Allosteric enhancers/modulators bind adenosine receptors at a different site than agonists and stabilize the tertiary complex between the agonist, receptor, and G-protein [49]. Positive allosteric modulators (PAMs) are better therapeutic options than orthosteric agonists because PAMs are tissue-specific, while native adenosine is easily degraded, and agonist adenosine does not readily penetrate the blood–brain barrier [50]. There are several potential ways to overcome adenosine- and AR-related obstacles, including the use of partial agonists, indirect receptor targeting, allosteric enhancers, prodrugs, nonreceptor-mediated effects, neoreceptors, and conditional knockouts [49].

### 3.1. A_1_ Adenosine Receptor (A_1_AR)

A_1_AR is a glycoprotein containing a single complex carbohydrate chain [51] that was cloned from cows, dogs, rabbits, guinea pigs, and chicks [11]. The clones encode a protein of 326 amino acids with a mass of approximately 36.7 kDa [12]. A_1_AR binds with G_i-1_, G_i-2_, G_i-3_, and G_o_ but not with G_s_ or G_z_, leading to many cellular responses, such as stimulation of K^+^ conductance, inhibition of Ca^2+^ conductance through N-type channels, stimulation of phospholipase C (PLC), and generation of Ca^2+^, protein kinase C (PKC) [8] and phosphoinositide 3 kinase (PI3K)/mitogen-activated protein kinase (MAPK) [52].

These receptors are distributed predominantly in the central nervous system (CNS), spinal cord, testis, kidney, and adipose tissue, whereas fewer receptors are distributed in the lung and pancreas [11]. They act on many effector or “second messenger” systems; for example, they inhibit catecholamines and histamine-stimulated AC in atrial and ventricular muscle cells of the heart [13].

All synthetic A_1_AR agonists are divided into specific types, i.e., A_1_AR agonists with an N^6^ substitution (N^6^-cyloalkyl-, N^6^-aylalkyl-, and N^6^-heterocyclic-substituted adenosine derivatives); C^2^ substitution, a variation of the ribose moiety (N-methanocarba analogues); and C1’-methyl and C2’-methyl substitutions, as well as AMP-579 [53].

### 3.2. A_2A_ Adenosine Receptor (A_2A_AR)

The majority of A_2A_ARs are distributed in the liver, heart, lung, immune systems (spleen, thymus, leucocytes, and blood platelets), and CNS [54,55]. They are coexpressed with D_2_ dopamine receptors (D_2_Rs) in γ-aminobutyric acid (GABA) striatopallidal neurons [56] but not active in striatonigral neurons that express the protachykinin and D_1_R genes [57]. A_2A_AR has been cloned from several species, including dogs, rats, humans, guinea pigs, and mice, and shows a high degree of homology among humans, mice, and rats [58]. A_2A_AR is a larger protein of 410–412 amino acids in length and 45 kDa [12].

A_2A_AR binding with G_s_ to activate AC results in the activation of cAMP-dependent protein kinase A (PKA) and PKC, which phosphorylates and activates various receptors, ion channels, phosphodiesterases, and phosphoproteins, such as cAMP-response element binding protein (CREB) and dopamine- and cyclic AMP-regulated phosphoprotein 32 (DARPP-32) [59]. These proteins interact with G_olf_ in the striatum of the basal ganglia because G_olf_ is highly expressed in the caudate-putamen, nucleus accumbens, and olfactory tubercle [60].

Five types of agonist A_2A_ARs exist: ribose-modified adenosine derivatives, purine-modified adenosine derivatives, ribose and purine-modified adenosine derivatives, partial agonists, and agonist radioligands [61]. Inversely, their antagonists are split into xanthine and nonxanthine derivatives (adenine derivatives and related heterocyclic compounds, simplified heterocyclic compounds unrelated to adenine or xanthine, and antagonist radioligands) [62].

### 3.3. A_2B_ Adenosine Receptor (A_2B_AR)

A_2B_AR has been cloned from the rat hypothalamus, human hippocampus, and mouse mast cells and is distributed in the peripheral organs, such as the bowel, bladder, lung, and vas deferens [59]. It interacts with G_s_ to induce the PKA signaling to increase cAMP and G_q11_-mediated activation of PLC to increase the levels of 1,4,5-inositol triphosphate (IP3)/diacylglycerol (DAG), activate PKC, and elevate intracellular Ca^2+^ levels [63,64].

A_2B_ARs are low-affinity receptors, and their affinity can be increased by interaction with PKC; therefore, the plasticity and versatility of A_2B_ARs make them potential triggers of signaling in multiple signaling cascades in many physiological responses [65].

In contrast with other ARs, A_2B_ARs have an antagonistic effect, which makes them interesting as therapeutic targets. Kalla RV et al. categorized their antagonists as xanthine-based, deazaxanthine-based, adenine-based, 2-aminopyridine-based, bipyrimidine-based, pyrimidone-based, imidazopyridine-based, pyrazine-based, and pyrazolo-triazolo-pyrimidine-based antagonists [64].

### 3.4. A_3_ Adenosine Receptor (A_3_AR)

The A_3_AR subtype was found after Zhou et al. found several cDNA sequences, one of which (R226) was highly expressed in the testis and less distributed in the lung, kidney, heart, and some parts of the CNS [66]. This confirmed a previous study that identified a cDNA encoding a novel GPCR in the rat brain [67]. A_3_AR has been cloned from rats, rabbits, sheep, and humans [68]. The structures of A_3_ARs from these organisms show some sequence homology, with a sequence homology of only 74% in rats versus sheep and humans and 85% between sheep and humans, which indicates significant interspecies differences in ligand recognition [69]. Unlike rats, human A_3_ARs are widespread, and the most abundant expression is found in the lung and liver [70].

A_3_AR coupling to G_i_ proteins inhibits AC and decreases cAMP accumulation and PKA activity, and A_3_ARs also bind with G_q_ proteins to stimulate PLC, resulting in increased Ca^2+^ levels and modulation of PKC activity [71]. In addition, they may activate the PLC pathway through the β, γ subunit [72].

A_3_AR agonists are classified as adenosine derivatives and xanthine-7-riboside derivatives [73]. In detail, structural manipulations of the A_3_AR agonist include *N^6^-*, *C^2^-*, and *5^′^*-substitutions, or suitable combinations of these, followed by modifications involving nonadenine nucleosides and nonnucleosides [71].

## 4. Adenosine Receptors and Diseases

### 4.1. A_1_AR

#### 4.1.1. A_1_AR in Inflammation

Activation of A_1_AR will generate both proinflammatory and anti-inflammatory responses [23]. By occupying A_1_AR, low concentrations of adenosine induce neutrophil chemotaxis and adherence to the endothelium [74] and upregulate endothelial P-selectin expression [75]. Interestingly, a high concentration of adenosine will induce antiadhesive effects via A_2_AR [76]. A_1_AR increases NK activity, induces O_2_^−^ generation from eosinophils, and induces the release of thromboxane A2 and IL-6 from endothelial cells. In addition, it also induces chemotaxis of dendritic cells (DCs), suppresses vesicular major histocompatibility complex (MHC) class I cross-presentation, and increases endothelial permeability [23].

The proinflammatory effects of A_1_AR in monocytes include enhancing Fcγ receptor-mediated phagocytosis [77], inducing secretion of vascular endothelial growth factor (VEGF) [78], and promoting monocyte differentiation into osteoclasts [79]. Indeed, inflammation induced by an agonist of A_2_AR, including secretion of IL-18 mediators such as IL-12, interferon (IFN)-γ, and tumor necrosis factor (TNF)-α, was abrogated by an agonist of A_1_AR [80].

In contrast, A_1_AR is a potent anti-inflammatory mediator in various kidney, heart, liver, renal, lung, and brain injury models [59]. A_1_AR protected mice from septic peritonitis by attenuating the hyperacute inflammatory response [81] and prevented the worsening of murine liver ischemia-reperfusion (IR) injury by reducing necrotic and apoptotic cell death [82]. In addition, A_1_AR protected against kidney injury via AKT activation [83]. The different functions of A_1_AR during inflammation might be determined by the model of study; in addition, with deletion of A_1_AR, other ARs might provide protection [23].

#### 4.1.2. A_1_AR in the Respiratory Systems

Adenosine regulates both proinflammatory responses and protection against lung injury through A_1_AR [14]. The distribution of A_1_AR in airway organs was low; however, its expression was found in smooth muscle tissues of asthmatic patients [84]. In fact, the expression of A_1_AR was upregulated in the airways of both animal models of allergic airway inflammation and human patients with asthma [85]. Specifically, A_1_AR induced bronchoconstriction and mucus production, evoking proinflammatory functions of monocytes and neutrophils and inducing vascular permeability [86], and might influence the severity of pulmonary inflammation and remodeling in chronic lung diseases [14].

Numerous studies have implicated a role for A_1_AR in respiratory diseases (Table 1). A novel A_1_AR antagonist, L-97-1, blocks allergic responses to house dust mites in an allergic rabbit model of asthma [87]. Regardless of the discontinuation of its study, EPI-2010, an antisense oligonucleotide targeting the A_1_AR promoter region, was safe and well tolerated in patients with mild asthma [88].

In contrast, A_1_AR also mediated anti-inflammatory effects via macrophages in ADA-deficient mice [89] and reduced polymorphonuclear (PMN) infiltration by inhibiting the release of chemotactic cytokines and decreasing microvascular permeability in lipopolysaccharide-induced lung injury [90]. Recently, CCPA, an A_1_AR agonist, decreased inflammation, edema, and neutrophil chemotaxis [91]. In addition, A_1_AR promoted the recruitment of leukocytes to the infected lung and attenuated lung injury [92].

#### 4.1.3. A_1_AR in the Cardiovascular Systems

A_1_AR protects the heart by downgrading inotropic effects. It binds with G_i_ to inhibit catecholamine-stimulated AC [93]; limit the action of β1 receptors stimulated by G_s_ cycling [94] via the involvement of PLC, PKC-ε, and p38 MAPK; and exclude heat shock proteins (HSP27) [95]. In addition, A_1_AR also inhibited norepinephrine release in the rat heart [96], providing protection in the postischemic setting [97].

Similarly, A_1_AR causes coronary contraction through PLC pathways [98] and increases soluble epoxide hydrolase (sEH) and cytochrome P450 A (CYP4A) [99] to protect the heart. In addition, A_1_AR activated the release of VEGF from monocytes [78] via the involvement of the extracellular signal-regulated kinase (ERK), c-Jun N-terminal kinase (JNK), and PI3K/AKT pathways [100]. A_1_AR also regulates myocardial substrate metabolism by decreasing plasma concentrations of insulin, glucose, and lactate [101], and inhibiting lipolysis [102]. Via cooperative interaction with both A_2A_AR and A_2B_AR, A_1_AR inhibits the necrosis cardiac cell ischemia model [103] through phosphorylation of ERK1/2 [104].

It is assumed that all ARs are involved in cardio hypertrophy and neovascularization [105]. A selective agonist of A_1_AR, CPA, prevented cardiac hypertrophy and heart failure (HF) in the left ventricular pressure-overload model [106]. An A_1_AR antagonist, BG9928, was well tolerated and significantly increased sodium excretion in patients with stable HF [107]. Conversely, rolofylline failed to protect renal function in acute heart failure (AHF) patients with volume overload and renal dysfunction [108].

Capadenoson, a new oral A_1_AR agonist, decreased the exercising heart rate at a comparable maximum workload in male patients with stable angina [121]. Moreover, this agonist also stimulated A_2B_AR, promoting cardioprotection and modulating cardiac fibrosis in heart disease [122]. Unfortunately, the clinical trial of capadenoson was discontinued [123]. A clinical trial using the partial agonist neladenoson (BAY 1067197) for the treatment of heart failure was safe without atrioventricular conduction disorders or neurological adverse effects [124]. A multiple-dose study of neladenoson in heart failure is still ongoing [123].

Taken together, A_1_AR activation ahead of ischemia may augment chemotaxis and neutrophil-dependent injury in cardioprotection; however, the effects of A_1_AR agonists during reperfusion and within both ischemic preconditioning and postconditioning settings remain controversial [125,126].

A_1_AR mediates negative chronotropy and dromotropy through inactivation of the inwardly rectifying K^+^ current (I_K,Ado_ or I_K,Ach_), inhibition of the inward Ca^2+^ current (I_Ca_), and activation of nitric oxide synthase (NOS) [105]. Furthermore, A_1_AR displays both antiarrhythmic and proarrhythmic effects [105]. High expression of A_1_AR results in sinus and AV node dysfunction and supraventricular arrhythmias [127].

At present, several full A_1_AR agonists have been used in clinical trials as antiarrhythmic agents. An optimal dose of a new A_1_AR selective agonist, tecadenoson (CVT-510), was successful in the conversion of paroxysmal supraventricular tachycardia (PSVT) into sinus rhythm [128], but its development for PSVT and atrial fibrillation (AF) was discontinued in 2009 [123]. Other agonists, such as selodenoson and PJ-875, have started in early phase evaluations for the treatment of AF [129]. Currently, to avoid the minor global effects of full agonists, including adverse reactions, desensitization, and arrhythmia (bradyarrhythmia or AF), partial A_1_AR agonists, such as CVT-2759, have been developed [130].

#### 4.1.4. A_1_AR in CNS

A_1_AR contributes to neuroprotection and is also involved in neurodegeneration. In fact, activation of A_1_AR under hypoxia leads to inhibition of presynaptic Ca^2+^ influx-related release of transmitters [131] such as dopamine, acetylcholine, GABA, and, especially, glutamate, to generate neuroprotection [132]. Moreover, A_1_AR also regulates potassium current, leading to hyperpolarization of the resting membrane potential mediated via G protein-dependent activation of inwardly rectifying K^+^ channels (GIRKs), activating PLC, and inhibiting AC [133].

Conversely, the levels of glutamate and *N*-methyl-d-aspartate (NMDA), which are responsible for neuronal damage, may increase during hypoxia and ischemia [134]. It was assumed that prolonged activation of A_1_AR increased A_2A_AR expression, which generated global damage and neurodegeneration in ischemic stroke [135]. Recently, an A_1_AR selective agonist, NNC-21-0136, was designed for neuroprotection in stroke models [123].

By binding with A_1_AR, adenosine reduced neuronal activity and pain in the spinal cord and periphery by inhibiting the cAMP, PKA, and Ca^2+^ channels, activating K^+^ currents, and interacting with the PLC, IP3, DAG, and β-arrestin pathways [136]. In preclinical studies, it seems that A_1_AR was dominant relative to A_2A_AR in inhibiting nociceptive input in the dorsal spinal cord [137]. The selective A_1_AR agonist GR79236X significantly relieves dental pain [138].

Moreover, Cl-ENBA, a potent and highly selective A_1_AR agonist, amplified nociceptive thresholds in spinal glial, and microglial changes occurred in neuropathic pain [139]. MRS7469 also plays a role in pain relief or inhibition of lipolysis [140]. Further development of other agonists, such as SDZ WAG 994, GR79236, and GW-493838, as well as the allosteric enhancer T62, was discontinued [141].

The activation of A_1_AR is crucial for keeping epileptic foci localized [142]; therefore, A_1_AR may be involved in the convulsive effect of diazepam, phenobarbital, or valproate in experimental seizure models [143]. Indeed, upregulation of A_1_AR suppressed seizure activity, which spread within the temporal lobe in rats [144], and could powerfully and bidirectionally regulate seizure activity [145]. An antagonist, DPCPX, inhibited the ketone diet-induced seizure effect [146].

The inhibition of A_1_AR during pathophysiological conditions (noxious stimulation and oxygen deficiency) causes mice to be more anxious [147] and aggressive [148]. In addition, an agonist of A_1_AR reduced the anxiogenic effects during ethanol withdrawal [149]. In overcoming the limitations of A_1_AR agonists, positive allosteric modulators, such as TRR469, offer a more physiological way to treat anxiety [150]. In addition, MRS5474, a potent selective A_1_AR agonist, has been developed as an antidepressant drug that is without cardiovascular side effects [151].

Relating to sleep and level of arousal, A_1_AR was shown to control the response of the circadian clock to light [152] and regulate homeostatic sleep after prolonged wakefulness [153] via inhibition of the ascending cholinergic neurons of the basal forebrain [154]. CPA inhibited the histaminergic system and promoted non-rapid eye movement (NREM) sleep [155].

In line with the fact that synaptic plasticity is the basis for learning and memory in different brain areas [156], A_1_AR attenuated long-term potentiation [157] and long-term depression and depotentiation in the hippocampus [158]. The mixed dual A_1_AR and A_2A_AR antagonist ASP5854 ameliorated motor impairments through neuroprotection (A_2A_AR antagonism) and enhanced cognitive function (A_1_AR antagonism) [159].

Via both A_1_AR and A_2A_AR, adenosine influences the two neurotransmitter systems most affected by schizophrenia—the glutamatergic and dopaminergic neurotransmission systems [160]. While A_1_AR inhibits D_1_R for dopamine, A_2A_AR reduces D_2_R recognition, coupling, and signaling [161]. CPA regulates prepulse inhibition (PPI), whereas PIA/L-PIA and 2-CLA/CHA are involved in memory functions and hyperlocomotion, respectively [162].

A_1_AR decreases tremor and controls the spread of excitability, thereby reducing the side effects of deep brain stimulation in Parkinson’s disease (PD) [163]. However, there is little information about the clinical study of A_1_AR in PD.

Drugs related to A_1_AR prevented the development of hindlimb dystonia in Huntington’s disease (HD). Several agonists, such as CPA, CCPA, CHA, and ADAC, have been developed; however, there are physiological limitations (capacities to cross the blood–brain barrier and adverse effects) to the use of A_1_AR-targeted drugs [162].

#### 4.1.5. A_1_AR in Metabolic and other Diseases

As mentioned earlier, A_1_AR agonists have potential use as antilipolytic agents. Indeed, inhibition of A_1_AR was associated with the enhancement of glucose-stimulated insulin, reduction of oxidative stress, and inhibition of the breakdown of triglycerides to free fatty acids (FFAs) [164].

The full A_1_AR agonists GR79236 and ARA are under development for the treatment of type 2 diabetes mellitus (T2DM), and their mechanisms of action includes reducing nonesterified fatty acids (NEFAs) and triglycerides (TGs) [165]. Unfortunately, the cardiovascular effects of full agonists raise a new problem; therefore, partial agonists that are capable of eliciting a greater effect in adipocytes than in the heart have been developed [166]. The A_1_AR partial agonist GS-9667/CVT-3619 was well tolerated in patients with T2DM and dyslipidemia, in whom it reduced plasma FFAs [167].

In addition, tecadenoson and BW-1433, an A_1_AR antagonist and A_2B_AR agonist, respectively, may lower FFA levels and improve glucose tolerance [168]. While A_1_AR selective agonists, such as SDZ WAG994 and RPR749, are under development for antilipolytic and hyperlipidemia, the development of other A_1_AR agonists, including trabodenoson and metrifudil 2 for glaucoma and glomerulonephritis, respectively, was discontinued for safety reasons [123].

### 4.2. A_2A_AR

#### 4.2.1. A_2A_AR in Inflammation

A_2A_ARs function as the most dominant anti-inflammatory effectors of extracellular adenosine through their expression on monocytes/macrophages, dendritic cells, mast cells, neutrophils, endothelial cells, eosinophils, epithelial cells, lymphocytes, NK cells, and NKT cells [169].

Activation of A_2A_ARs inhibits neutrophil adhesion to endothelial cells, formation of reactive oxygen species, and adherence of *N*-formyl methionyl-leucyl-phenylalanine (fMLP)-activated neutrophils to the endothelium and downregulates endothelial cell surface proteins, including Mac-1, β2-integrin, L-selectin, vascular cell adhesion molecule-1 (VCAM-1), intracellular adhesion molecule-1 (ICAM-1), alpha 4/beta 1 integrin VLA4, and platelet cell adhesion molecule. Additionally, it decreases TNF-α, macrophage inflammatory protein (MIP)-1α/CCL3, MIP-1β/CCL4, MIP-2α/CXCL2, MIP-3α/CCL20, leukotriene LTB4, and platelet-activating factor (PAF) [23].

During innate immunity, activation of A_2A_AR on adaptive immune cells shifts Th1 to Th2 responses by suppressing IL-12 and increasing IL-10 [170]. In parallel, secretion of TNF-α, IL-6, and IL-8 is also diminished [171]. Interestingly, the decreases in IL-10 and IL-6 under A_2A_R blockade and in knockout (KO) mice resulted in an increase in survival from polymicrobial sepsis [172]. In lymphocytes, A_2A_R also inhibited the production of IFN-γ, IL-4, and IL-2 [173].

Furthermore, the prodrug 2-(cyclohexylethylthio)adenosine 5’-monophosphate (chet-AMP) delivered potent immunosuppression with negligible vasodilatory activity in experimental rheumatoid arthritis (RA) models [174]. In addition, a deoxyadenosine derivative, polydeoxyribonucleotide (PDRN), acted on A_2A_AR in relieving pain, improving the clinical signs of arthritis, and reducing histological damage in an arthritis model [175].

In contrast, other agonists resulted in poor outcomes. Regadenoson 21 (CVT-3146) for sickle cell disease was not able to produce a statistically significant reduction or clinical efficacy [176]. Clinical trials of sonedenoson (MRE-0094) and spongosine (BVT.115959) for diabetic foot ulcers and diabetic neuropathic pain, respectively, were discontinued [123].

#### 4.2.2. A_2A_AR in the Respiratory System

The expression of A_2A_AR in the lung is distributed widely, including expression on resident macrophages, bronchial epithelial cells, mast cells, eosinophils, neutrophils, and lymphocytes [169]. Activation of A_2A_AR by adenosine binding affects multiple aspects of the inflammatory process, modulating neutrophil activation and degranulation, oxidative species production, adhesion molecule expression, cytokine release, and mast cell degranulation; therefore, selective A_2A_AR agonists have an important role in airway inflammation and neutrophil–monocyte-mediated lung tissue injury [14].

In asthmatic airways, A_2A_AR suppresses inflammation by reducing neutrophil adherence to the endothelium and inhibiting the fMLP-induced oxidative burst, superoxide anion generation, and LPS-induced TNF-α expression [85]. Moreover, it also inhibits histamine and tryptase release but stimulates wound healing and VEGF secretion [86]. Adversely, angiogenesis, as an important characteristic of airway remodeling in human asthma, may limit the development of such anti-asthma drugs [85].

A_2A_AR agonists in animal models of asthma and chronic obstructive pulmonary diseases (COPD) have resulted in good outcomes, yet subsequent clinical trials in humans have not been successful. CGS-21680 has anti-inflammatory activity in a model of allergic asthma in the Brown Norway rat [109], and UK371,104 inhibited capsaicin-induced bronchoconstriction [110]. However, an inhaled A_2A_AR agonist, GW328267X, in patients with nonsmoking asthma was unsuccessful in protecting against the allergen-induced early and late asthmatic reactions [111]; hence, its study and the study of UK371,104 were discontinued due to limited efficacy [177].

Recently, regadenoson has been demonstrated to be safe to use in patients with mild to moderate COPD and asthma [112]. Another agonist, apadenoson, is still in a phase I trial for asthma and COPD [59].

The selective agonist CGS-21680 also protected the lung against shock-induced injury [113]. In addition, pentoxifylline showed anti-inflammatory effects in LPS-induced lung injury through an A_2A_AR-dependent pathway [114].

#### 4.2.3. A_2A_AR in the Cardiovascular Systems

In contrast to A_1_AR, activation of A_2A_AR prior to ischemia is not effective. Administration during or prior to reperfusion minimizes myocardial ischemia-reperfusion injury; hence, A_2A_AR works in ischemic postconditioning [126]. As an anti-inflammatory molecule, A_2A_AR protects the heart through inhibition of leukocyte-dependent inflammatory processes, angiogenesis, enhanced coronary vasodilatation during reperfusion, modified myocardial contraction [178], and increased inotropy through transient Ca^2+^ augmentation via a cAMP/PKA-dependent mechanism [179].

The selective A_2a_AR agonists CGS 21680C and ATL-193/ATL-146e protected tissue during ischemia-reperfusion injury by reducing myocardial infarct size without elevating coronary blood flow [180,181]. However, A_2A_AR not only protects against ischemia but also increases the occurrence of cardiac arrhythmias [182]. In fact, A_2a_AR indirectly alters contractility by modulating the A_1_AR antiadrenergic effect rather than increasing cardiac contractility directly [183]. A new study confirmed that the A_2a_AR agonist LASSBio-294 might be an alternative treatment for heart failure due to ischemia or hypertension [184].

Adenosine controls coronary blood flow regulation through A_2a_AR in both coronary smooth muscles and coronary endothelial cells [185] in a mechanisms that involves several second messengers and effectors, including p38-MAPK, IP3, NO, and K^+^ channels [105]. These findings led to the development of various A_2a_AR agonists for both diagnostic (myocardial perfusion imaging (MPI)) and therapeutic interventions. Regadenoson (CVT-3146), binodenoson (MRE-0470/WRC-0470), evodenoson (ATL-313, DE-112), sonedenoson (MRE-0094), and apadenoson (ATL-146e) have been approved by the FDA for use in MPI [123,185].

Along with being cardioprotective via inhibition of the inflammatory response, A_2A_AR agonists (PSB-15826, PSB-12404, and PSB-16301) also modulate platelet aggregation [186], and ATL313 regulates cholesterol homeostasis [187]. Perhaps the pharmaceutical industry will focus on the anti-inflammatory effects of A_2A_AR agonists against excess cholesterol accumulation to develop new cardiovascular therapies [188]. A2A receptor agonists are the focus of efforts by the pharmaceutical industry to develop new cardiovascular therapies, and pharmacological actions of the atheroprotective and anti-inflammatory drug methotrexate are mediated via release of adenosine and activation of the A2A receptor.

A list of AR ligands currently undergoing clinical trials as novel therapeutic treatments and their effects on cardiovascular and metabolic diseases is presented in Table 2.

#### 4.2.4. A_2A_AR in the CNS

According to physiological factors, neuromodulation of adenosine through activation of high-affinity A_2A_AR is important [163]. Indeed, A_2A_AR was associated with neurodegeneration due to its excitatory effects [135]. Presynaptic A_2A_AR counteracts the inhibitory effect of presynaptic A_1_AR on glutamate release from axon terminals, inducing glutamate release that would predispose to excitotoxic injury, yet A_2A_AR may also exert a neuroprotective effect by promoting vasodilation and preserving cerebral blood flow autoregulation [143]. Recently, SCH58261, an A_2A_AR antagonist, has proven efficacious in halting injury by inhibiting pERK1/2 in hippocampal neurons [200].

Similarly, A_2A_AR and A_1_AR of hippocampal CA1 have the opposite effect in spreading piriform cortex kindled seizures, in which A_2A_AR tends to have a convulsive effect [201]. In fact, the A_2A_AR antagonist ZM241385 had a potent anticonvulsant profile with few adverse effects [202] and suppressed subsequent recovery sleep [203]. Concerning sleep and arousal, the A_2A_AR agonist CGS21680 promoted both rapid eye movement (REM) and NREM sleep [133].

Activation of A_2A_AR increased the release of the potent anti-inflammatory cytokine IL-10, leading to suppression of pain [61]. CGS21680 produced a long-duration reversal of mechanical allodynia and thermal hyperalgesia [155]. Another agonist, spongosine (BVT.115959, CBT-1008), was effective in a clinical trial for diabetic neuropathic pain, but the production was discontinued [204].

A_2A_AR KO mice show increased anxiety [57], and there was an association between a polymorphism in the A_2A_AR gene and panic disorder [205]. In contrast, A_2A_AR KO mice were less sensitive to “depressant” challenges [206], which might be linked to interactions with dopaminergic transmission in the frontal cortex [207]. Recently, istradefylline became the first therapeutic agent for depression and anxiety [208].

Apart from D_2_R, recent studies also confirmed the involvement of the interaction between A_2A_AR/D3 and A_2A_AR/mGlu5 in schizophrenia [209]. Therefore, multiple adenosinergic targets, including A_1_R and A_2A_R, have potential as therapeutic targets for schizophrenia [210].

HD is characterized by a decrease in A_2A_R due to neurodegeneration of the GABA/enkephalin striatopallidal neurons [134]. Interestingly, both A_2A_R agonists and antagonists have shown beneficial effects [211]; thus, the potential exploitation of A_2A_R ligands in HD is still contentious, reflecting the complexity of A_2A_R regulation in this disease [132]. As in HD, there was a reduction in A_2A_R in regions of high density, such as the striatum, in Alzheimer’s disease (AD) [132]. Recently, the selective A_2A_R antagonist MSX-3 prevented the development of memory deficits in an AD mouse model without altering hippocampal and cortical gene expression [212].

A key finding in PD is the colocalization and reciprocal antagonistic interactions between A_2A_R and D_2_R [134]; hence, A_2A_R antagonists may not only relieve motor deficits in PD but also potentially prevent the degeneration of dopaminergic mesencephalic neurons [213]. Some PD drugs are still under development, including tozadenant/SYN115, DT1133, ZM241385, ST1535, and istradefylline; however, other drugs (preladenant, ASP5854, and vipadenant) were discontinued [61].

A_2A_R interacts antagonistically with D_2_R to counteract drug addiction-induced behavioral effects [214]. The high density of presynaptic and postsynaptic A_2A_Rs that regulate glutamatergic transmission in the brain leads to the possibility of A_2A_Rs as new therapeutic agents for drug addiction [163]. The adenosine agonists NECA and CGS 21680 inhibit cocaine-seeking behaviors [215].

The effects of drugs targeting A_2A_AR and other ARs that are currently undergoing clinical trials as novel therapeutic treatments in CNS diseases on the CNS system are listed in Table 3.

### 4.3. A_2B_AR

#### 4.3.1. A_2B_AR in Inflammation

A_2B_AR is expressed on most inflammatory cells and has both proinflammatory and anti-inflammatory effects [177]. It generates anti-inflammatory effects by coupling with protein G_s_ and proinflammatory effects by coupling with protein G_q_ [216]. Due to its low affinity, A_2B_AR might require high concentrations of adenosine, for instance, during pathological conditions, to be significantly activated [217].

The proinflammatory effects of A_2B_AR included the induced secretion of IL-6 from macrophages and IL-1β, IL-13, IL-3, IL-8, IL-4, and VEGF from mast cells [218]. Moreover, A_2B_AR induced the production of IL-19 and TNF-α from bronchial epithelial cells [219].

A_2B_AR mediates anti-inflammatory effects in neutrophils by inhibiting adhesion to endothelial cells [23], preventing the production of TNF-α and IL-1β, as well as macrophage proliferation, and stimulating IL-10 secretion from macrophages [216]. Interestingly, A_2B_AR has proinflammatory and anti-inflammatory actions in mouse bone marrow-derived mast cells (BMMCs) [225].

Adenosine regulates bone metabolism and wound healing. Deletion of A_2B_AR inhibits periosteal development with subsequent consequences on endochondral ossification and growth plate regulation [226]. In fact, it modulated osteoblast differentiation [227]. Moreover, these receptors also regulate myocardial repair and remodeling by delaying the deactivation of myofibroblasts [228].

Receptor inhibition or knockout of A_2B_AR in mice suppressed intestinal inflammation and attenuated disease in murine colitis/inflammatory bowel diseases [63]. A_2B_AR in epithelial cells appeared to attenuate colonic inflammation through a specific barrier repair response, namely, phosphorylation of vasodilator-stimulated phosphoprotein [136]. Instead, the deletion of A_2B_AR or inhibition by PSB1115 exacerbated the acute inflammatory phase of dextran sodium sulfate colitis [229].

Consistent with their anti-inflammatory effects, CVT-6883 and MRS-1754 diminished the clinical symptoms of experimental autoimmune encephalitis and protected the CNS from immune damage [230]. Administration of the antagonist BAY 60-6583 for four weeks in a high-fat diet mouse model restored endocrine function and reduced inflammation [123].

Taken together, the use of A_2B_AR agonists or antagonists for the treatment of the associated disease states is still controversial, and another important variable for agonists, i.e., signaling bias, needs to be considered [231]. Figure 2 illustrates the effects of AR on immune cells.

#### 4.3.2. A_2B_AR in the Respiratory System

A_2B_AR binds with both G_s_ and G_q_ proteins to mediate airway reactivity, inflammation, and remodeling in asthma [85]. Activation of A_2B_AR in mast cells induced the release of inflammatory mediators, leading to bronchoconstriction [232]; therefore, the use of antagonist of A_2B_AR is based on the cellular effects downstream of A_2B_AR mast cells [86].

A selective A_2B_AR antagonist, CVT-6883, inhibited airway reactivity and inflammation in a mouse model of asthma [116]. Other compounds, including CGS15493, WO-00125210, and ATL-907, were under development in phase I and phase II trials for the treatment of asthma [86].

The role of mast cells in COPD is controversial, but there is a significant inverse correlation between the binding parameters of A_2B_AR and the FEV1/FVC ratio [233]. Indeed, the combination of the selective A_2B_AR agonist/antagonist BAY 60-6583 and dexamethasone may have clinical applications in COPD by inducing genes with anti-inflammatory activity [115]. Furthermore, some studies confirmed that adenosine interacting with A_2B_AR served an essential role in the pathogenesis of COPD and pulmonary fibrosis [234] by stimulating the secretion of IL-6 and the differentiation of pulmonary fibroblasts into myofibroblasts [235].

Administration of the A_2B_R antagonist GS-6201 or deletion of A_2B_R attenuated vascular remodeling and hypertension associated with interstitial lung disease [118]. Additionally, an A_2B_R antagonist neutralized the increase in metalloproteases and inhibitors of proteases, tissue inhibitor of metalloproteinase-1 (TIMP-1), matrix metalloproteinase (MMP)-9, and MMP-12 [119].

A_2B_AR is also a potential target in acute lung injury (ALI). In a ventilator-induced lung injury (VILI) mouse model, the deletion of the A_2B_AR gene was associated with reduced survival time and increased pulmonary albumin leakage [117]. Recently, aerosolized BAY 60-6583 was shown to attenuate pulmonary edema, improve histologic lung injury, and diminish lung inflammation in ALI [236].

#### 4.3.3. A_2B_AR in Cardiovascular Systems

There is no “direct” evidence of the effect of A_2B_AR on ventricular myocytes [178]; however, A_2B_AR was shown to be required for A_1_AR-mediated cardioprotection [237]. In fact, adenosine levels during myocardial ischemia-reperfusion increase; therefore, low-affinity A_2B_AR still has cardioprotective effects upon its activation [238].

The results of studies on ischemia are conflicting. A_2B_AR plays an important role in ischemia preconditioning (IPC) [63]. Indeed, A_2B_AR attenuated myocardial infarction via hypoxia-inducible factor (HIF)-1α and the circadian rhythm protein Per2 [239]. In contrast, a previous study reported that A_2B_AR was not involved in the early phase of IPC but might contribute to the later stages by inducing stress-responsive genes [240]. Recently, GS-6201 was shown to attenuate cardiac remodeling by reducing caspase-1 activity after acute myocardial infarction [191], and BAY 60-6583 protected the heart against myocardial IR injury by modulating macrophage phenotype switching via the PI3K/AKT pathway [190].

A_2B_AR may inhibit smooth muscle cell-associated hypertension [241]. In fact, KO mice have normal blood pressure [242]. Recently, the deletion of A_2B_AR was shown to protect against salt-induced hypertension and stroke [243] and inhibit increases in mean arterial blood pressure [244].

A_2B_AR may contribute to the pathogenesis of atherosclerosis. During hypoxia, A_2B_AR promoted HIF-1α-induced differentiation of macrophages and foam cells (FC) in preventing atherosclerosis plaque formation [245]. Administration of BAY 60-6853 altered the lipid profile and decreased atherosclerosis in mice [189].

#### 4.3.4. A_2B_AR in Metabolic Diseases

It is known that A_2B_AR has roles in glucose homeostasis and lipid metabolism, insulin secretion and resistance, inflammation, β-cell survival, and kidney protection [246]. The nonselective A_2B_AR agonist NECA, via immunomodulatory effects, inhibited diabetes and protected the pancreas [192]. Moreover, the antagonist ATL-801 increased insulin action in the liver and glucose uptake in skeletal muscle and brown adipose tissue [193].

In diabetic nephropathy, MRS-1754 inhibited high glucose via VEGF [194] and improved kidney tissue nitrite levels [247]. Tak et al. asserted that endothelial A_2B_AR signaling was implicated in protection from diabetic nephropathy [248].

#### 4.3.5. A_2B_AR in Cancer

A_2B_AR is highly expressed in tumor cells and promotes tumor cell proliferation; hence, it might serve as a target for novel therapies or combination therapies for cancer [249]. A_2B_AR promoted tumor-inducing M2 macrophages (tumor-associated macrophages); however, it had less effect on decreasing cancer cell proliferation and metastasis and inducing proapoptotic effects [250].

The A_2B_AR antagonist ATL801 repressed the growth of bladder and breast tumors and reduced the metastasis of breast cancer cells [251]. Another antagonist, PSB1115, inhibited the accumulation of tumor-infiltrating myeloid-derived suppressor cells (MDSCs) and restored an efficient antitumor T cell response in a mouse model melanoma [252]. In contrast, A_2B_AR promoted tumor growth through VEGF and IL-8 [253,254].

Taken together, these results shown that the roles of A_2B_AR in cancer are complex. Gessi et al. showed that A_2B_AR releases angiogenic factors promoting tumor growth; however, it may exert an inhibitory signal on tumor cell proliferation [255].

### 4.4. A_3_AR

#### 4.4.1. A_3_AR in Inflammation

The dual effect of A_3_AR in the inflammation process involves many cells that can have overlapping and opposing function [23]. Its presence in almost all inflammatory cells suggests their involvement in a number of inflammatory pathologies, spanning from wound healing and remodeling to lung injury, inflammatory bone loss, autoimmunity, and eye diseases [69].

A_3_AR produced a proinflammatory effect by inducing the release of histamines and other allergic mediators [256], preventing eosinophil chemotaxis [257], and inhibiting apoptosis [258]. A_3_AR also increased rapid inflammatory cell influx by attracting eosinophils and macrophages in the lung [259] and promoting maturation of DC and Ca^2+^ signaling [260].

On the other hand, A_3_AR inhibited LPS-stimulated release of TNF-α and nitric oxide (NO) via the NF-κβ, ERK1/2 and PI3K/AKT pathways [261,262]. Interestingly, both of the resulting modulators were found to increase neutrophil chemotaxis [263] and, in contrast, mediate inhibition of oxidative burst in human neutrophil and promyelocytic HL60 cells [264]. In addition, A_3_AR modulates lymphocyte T cell activation [265].

A_3_AR agonists suppressed the production of MIP-1α, collagen-induced arthritis [266], and TNF-α [267]. IB-MECA (CF101, piclidenoson) is safe and generally well tolerated in treating rheumatoid arthritis [268]. A recent clinical trial showed that LUF6000, an A_3_AR allosteric modulator, induced an anti-inflammatory effect in animal models of arthritis via deregulation of PI3K, IKK, IκB, Jak-2, and STAT-1 signaling, resulting in decreased levels of nuclear factor (NF)-κβ [269].

A_3_AR delivered significant protection from murine septic peritonitis primarily by attenuating the hyperacute inflammatory response in sepsis and IP lung injury [270,271]. Currently, IB-MECA is in a trial for patients with moderate to severe plaque psoriasis [272]. Furthermore, the antagonists A_3_AR, MRS 1292, and OT-7999 inhibited shrinkage of human nonpigmented ciliary epithelial cells and reduced mouse intraocular pressure (IOP) in an animal model of glaucoma [273,274]. A trial of IB-MECA for dry eyes was unsuccessful, yet IB-MECA was able to reduce IOP [275].

#### 4.4.2. A_3_AR in the Respiratory System

A_3_AR plays roles in asthma, COPD, lung fibrosis, and pulmonary inflammation. Adenosine-induced airway hyperresponsiveness (AHR) in mice occurs largely through A_3_AR in mast cells [276] via G_i_ and PI3K signaling pathway-mediated accumulation of intracellular Ca^2+^ [277]. While the role of rodent A_3_AR-activated mast cells in AHR is firmly established, the mechanism of A_3_AR-induced release of inflammatory mediators in humans remains elusive [278]. Most A_3_ARs are expressed in lung eosinophils instead of mast cells, which mediate the inhibition of both degranulation and O_2_^−^ release [257,279]. Indeed, A_3_AR has an important role in regulating lung eosinophilia and mucus production in an environment of elevated adenosine [280].

Nebulized IB-MECA directly induced lung mast cell degranulation while having no effect in A_3_AR KO mice [277]. Likewise, adenosine induces AHR indirectly by activating A_3_AR on mast cells [276]. Although A_3_AR is absent in human lung mast cells, it inhibits degranulation of eosinophils, so it may be useful in eosinophil-dependent allergic disorders, such as asthma and rhinitis [68]. In fact, a selective adenosine A_2B_AR agonist/A_3_AR antagonist had no significant effect on rhinorrhea, the number of sneezes, or peak nasal inspiratory flow measurements [281]. The combined A_2B_AR/A_3_AR antagonist QAF 805 failed to increase the PC20 [85].

In contrast, IB-MECA has an important role in IR-induced lung injury through the upregulation of phosphorylated ERK [271] and the nitric oxide synthase (NOS)-independent pathway [120]. Furthermore, the deletion of A_3_AR enhanced pulmonary inflammation by increasing eosinophil-related chemokines and cytokines [282].

Overall, A_3_AR is involved in both proinflammatory and anti-inflammatory responses depending on the cell type involved, although data in the recent literature appear to lean towards a protective effect [68].

#### 4.4.3. A_3_AR in Cardiovascular Systems

The cardioprotective effect of A_3_AR is similar to the combined effects of A_1_AR and A_2A_AR. Activation of the A_3_AR either prior to ischemia or during reperfusion was useful in cardioprotection [125]. Administration of IB-MECA before ischemia or during reperfusion effectively reduces infarct size [197], and CI-IB-MECA (CF102, namodenoson) inhibited myocardial ischemia-reperfusion injury [198]. Additionally, CP-532,903, a highly selective agonist of A_3_AR, protected against ischemia-reperfusion injury [199].

It is assumed that A_3_AR induces cardioprotection via PKC, PI3-kinase, ERK, mitochondrial K_ATP_ channels [178], and NO [283]. Activation of A_3_AR limited myocardial injury in an isolated rat heart and improved survival in isolated myocytes, possibly via antiapoptotic and antinecrotic mechanisms [284]. Indeed, the basic mechanism in A_3_AR-associated cardioprotection remains poorly defined and may vary between species (e.g., rodents vs. humans) and protective responses (e.g., acute vs. delayed protection) [285].

Paradoxically, low-level expression in the heart provides effective protection against ischemic injury without adverse effects, whereas higher levels lead to the development of a dilated cardiomyopathy [286]. Hinze et al. first reported that A_3_AR induces proliferation by activating PLC and inducing the transcription factors early growth response (EGR) 2 and EGR3 [287]. A_3_AR acts against the protective effect of adenosine in the overloaded heart; therefore, attenuation of A_3_AR might be a novel approach to treat pressure overload-induced left ventricular hypertrophy and dysfunction [288]. Moreover, deletion of A_3_AR protects the heart against renal and cardiovascular disease [289].

A_3_AR induces vasoconstriction via mast cell-produced histamine and thromboxane, inhibition of cAMP in smooth muscle and aorta, and enhancement of cellular antioxidants [71]. A_3_AR-mediated contraction through the endothelium may play a role in cardiovascular inflammation, including hypertension and atherosclerosis, by affecting the cyclooxygenase signaling pathways [290], and is also linked with reactive oxygen species (ROS) generation via activation of nicotinamide adenine dinucleotide phosphate (NADPH) oxidase/Nox2 [291]. A_3_AR is involved in atherosclerosis through modulation of the hyaluronic matrix [292]. Furthermore, it induces VEGF secretion and FC synthesis in a HIF-1α-dependent manner [293].

The latest study shows that LJ-1888, a selective antagonist for A_3_AR, is a feasible novel candidate for the treatment of atherosclerosis and hypercholesterolemia [196]. Moreover, LJ-2698, a highly selective and species-independent A_3_AR antagonist, ameliorated diabetic kidney complications [195].

The role of cardioprotection by A_3_AR is enticing, enigmatic, and elusive [294]. Further clinical studies on cardioprotection induced by adenosine receptors need to establish the role of adenosine receptor agonists and antagonists in more clinically relevant models of myocardial ischemia [238].

#### 4.4.4. A_3_AR in CNS Systems

The expression of A_3_AR in the brain is low and difficult to detect using sensitive in situ hybridization or receptor-labeling autoradiography methods [295], yet A3AR exists in neurotransmission [71].

Several studies have reported a neuroprotective function. Under ischemia, CA1 hippocampal A_3_AR might exert A_1_-like protective effects on neurotransmission, but severe ischemia would transform the A_3_ receptor-mediated effects from protective to injurious [296]. Administration of IB-MECA chronically prior to forebrain ischemia resulted in improved postischemic cerebral blood circulation, survival, and neuronal preservation, whereas a negative effect was found when given acutely [222].

In line with this, acute stimulation by CI-IB-MECA induced cell death in a concentration-dependent fashion by inhibiting cAMP production [297]. The timing of treatment also produced opposing results, as administration of IB-MECA prior to and after transient middle cerebral ischemia resulted in an increase and decrease in the infarct size, respectively [223]. The protective role of A_3_AR in ischemia is possibly mediated by the preservation of ischemia-sensitive microtubule-associated protein 2 (MAP-2), enhancement of the expression of glial fibrillary acidic protein [298], depression of NOS, stimulation of glial CCL2 synthesis [299], and delay of irreversible synaptic failure [300].

On the other hand, A_3_AR stimulated convulsion in the CA1 [301] and CA3 [302] regions of the rat hippocampus. A new antagonist, ANR235, increases the stability of I_GABA_ in different epileptic tissues and may offer therapeutic opportunities in human epilepsy [224]. In serotonin (5-HT)-linked brain diseases, A_3_AR is also involved in the rapid stimulation of presynaptic serotonin transport [303] mediated by protein kinase G_1_ and p38 MAPK [304]. A_3_AR might function as a potential candidate in brain inflammation treatment by suppressing TNF-α production [262]. Further studies are mandatory for establishing the neuroprotective role of A_3_AR before A_3_AR agonists or antagonists can be used clinically.

#### 4.4.5. A_3_AR in the Digestive and Renal Systems

A_3_AR delivers protection in inflammatory gastrointestinal diseases. IB-MECA treatment reduced oxidative damage and inflammatory mediators in the colon, including IL-1, IL-6, IL-12, MIP-1α, and MIP-2 [305]. This agonist also provides protection from CLP-induced mortality and acute organ dysfunction in murine septic peritonitis primarily by attenuating the hyperacute inflammatory response [270].

A recent study showed that A_3_AR had anti-inflammatory activity via inhibition of proinflammatory cytokine expression associated with the inhibition of NF-κβ signaling pathways in murine dextran sulfate sodium (DSS) colitis [306]. Blocking A_3_AR produced protection against ischemia- and myoglobinuria-induced renal failure [307]. Recently, selective A_3_AR antagonism attenuated renal ischemia-reperfusion injury [308].

#### 4.4.6. A_3_AR in Cancer

Primary and metastatic solid tumors showed high expression of A_3_AR [309], which might be related to overexpression of NF-κβ [310]. In addition, high expression has been observed using biochemical methods in many types of cancer cells, including astrocytoma, melanoma, lymphoma, sarcoma, glioblastoma and colon, liver, pancreatic, prostate, thyroid, lung, breast, and renal carcinomas [69]. Table 4 illustrates the A_3_AR and A_2B_AR ligands that are currently undergoing clinical trials as novel therapeutic treatments in cancer.

It has long been known that A_3_AR is involved in the regulation of the cell cycle, and both proapoptotic and antiapoptotic effects have been reported depending on the level of receptor activation [311]. In the event of tumor inhibition, adenosine plays a role in the resistance of muscle to tumor metastases [312], and its antiproliferative effect occurs mainly via A_3_AR-induced cell cycle arrest in the G_0_/G_1_ phase and decreases in telomeric signaling in these cells [313]. CI-IB-MECA protects the retinal ganglion by increasing survival [314]. Additionally, it mediates a tonic proliferative effect in colon tumor cells [315]. It is possible that the antiapoptotic effect is related to oxygen concentration [59].

Apart from hypoxia, HIF-1α has a role in cancer-associated angiogenesis [316]. A_3_AR upregulates HIF-1α protein expression in a dose-dependent and time-dependent manner via p44/p42 and p38 MAPK [317]. Moreover, an A_3_AR antagonist inhibited VEGF protein accumulation [318] and reduced tumor size and blood vessel formation under hypoxic conditions in glioblastoma cells [319].

IB-MECA inhibited tumor cell growth of HCT-116 human and CT-26 murine colon carcinoma cell [320] and B16-F10 melanoma cells [321]. Furthermore, other in vivo studies have shown that IB-MECA in PC-3 prostate carcinoma cells [322] and CI-IB-MECA in N1S1 rat hepatocellular carcinoma (HCC) cells [323] induced apoptosis and tumor growth inhibition via deregulation of the Wnt and NF-κβ signaling pathways.

The oral bioavailability of synthetic A_3_AR agonists renders them potentially useful in three different modes of treatment: as stand-alone anticancer treatments, in combination with chemotherapy to enhance its therapeutic index, and as agents inducing myeloprotection [324]. The latest study confirmed that combination use of A_3_AR agonists produces anticancer effects in an apoptosis-, autophagy-, and ROS-dependent manner [325].

## 5. Conclusions

Pharmacological interventions related to AR ligands as new drugs has been widely developed. Most ARs have roles as both proinflammatory and anti-inflammatory mediators as well as in organ protection and degeneration, and these opposing roles depend on the timing and concentration. As a consequence of being widely distributed throughout the body, adenosine activates protection for tissues and cells against injury, hypoxia, and ischemia; however, it also stimulates various adverse effects, especially in the cardiovascular and respiratory systems. Therefore, future studies should focus on the abundant amount of adenosine and reducing its harmful impact. Overall, focusing on partial agonist and allosteric enhancers might be the most appealing pharmacological angle for successful treatment in cancer, inflammation, cardiometabolic diseases, respiratory diseases, and neurovascular diseases.

## Figures and Tables

**Figure 1 cells-09-00785-f001:**
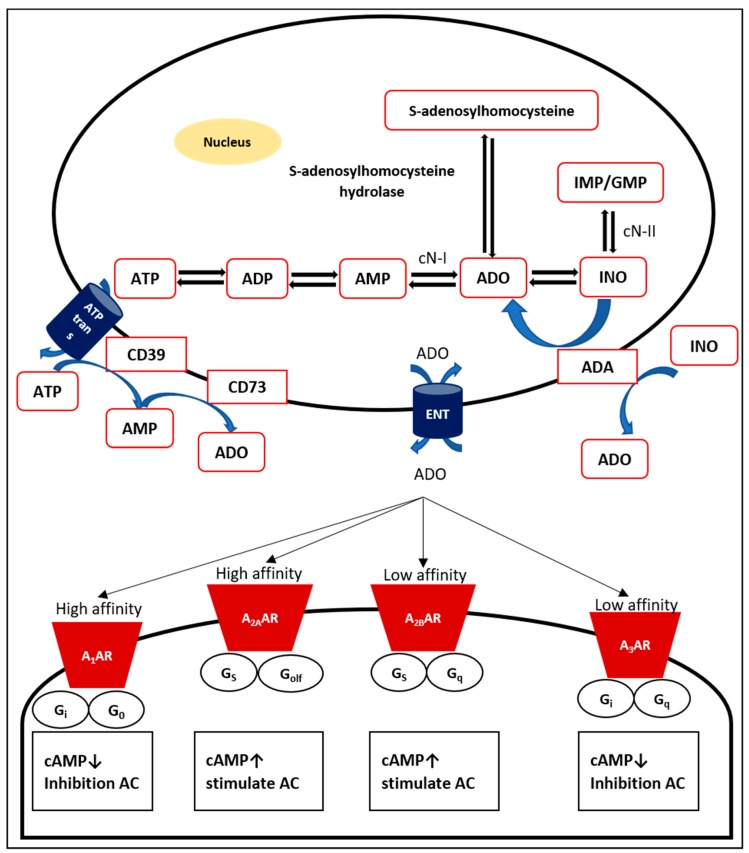
Production, transport and metabolism of adenosine. Intracellular adenosine is produced via dephosphorylation from the main source, AMP, via both cN I and cN-II and hydrolysis of S-adenosyl-homocysteine through the enzyme S-adenosyl-L-homocysteine hydrolase. The extracellular formation of adenosine is the result of enzymatic cascades consisting of ATP transport, hydrolysis of ATP and ADP by CD39 to form AMP, and dephosphorylation of AMP by CD73. Extracellular adenosine binds adenosine receptors (A_1_AR, A_2A_AR, A_2B_AR, and A_3_AR) on the surface of cells. Each AR is a GPCR that transmits the signal from adenosine by activating cAMP.

**Figure 2 cells-09-00785-f002:**
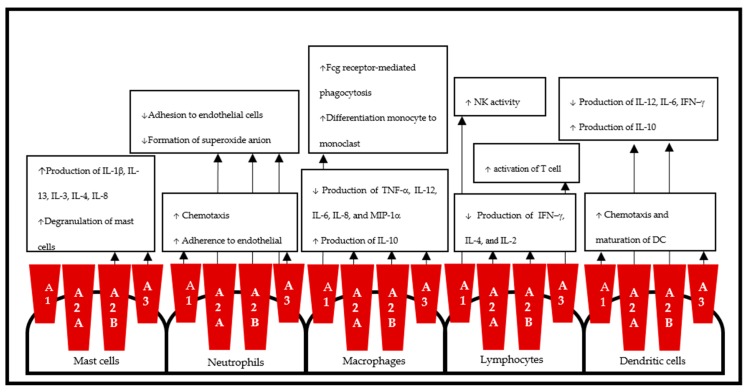
Schematic representation of adenosine receptors in immune cells. Adenosine receptors carry out the proinflammatory and anti-inflammatory effects of adenosine through predominant immune cells: macrophages, neutrophils, mast cells, lymphocytes, and dendritic cells.

**Table 1 cells-09-00785-t001:** Adenosine receptor drugs in respiratory diseases.

Receptors	Diseases	Drugs	
		Selective Full Agonist	Antagonist/Partial Agonist
**A_1_AR**	COPD and asthma		L-97-1 [87] EPI-2010 [88]
	Lung injury	CCPA [91]	
**A_2A_AR**	COPD and asthma	CGS-21680 [109] UK371,104 (DC) [110] GW328267X (DC) [111] Regadenoson [112] Apadenoson [59]	
	Lung injury	CGS-21680 [113] PTX [114]	
**A_2B_AR**	COPD and asthma	BAY 60-6853 [115]	CVT-6883 [116] CGS15493, WO-00125210, and ATL-907 [86]
	Lung injury	BAY 60-6583 [117]	
	Fibrosis and interstitial lung diseases	GS-6201 [118]	CVT-6883 [119] QAF 805 [85]
**A_3_AR**	COPD and asthma		QAF 805 [85]
	Lung injury	IB-MECA [120]	

DC = discontinued.

**Table 2 cells-09-00785-t002:** Adenosine receptor drugs in cardiovascular and metabolic diseases.

Receptors	Functions		Diseases	Drugs	
				Selective Full Agonist	Antagonist/Partial Agonist
**A_1_AR**	**Inotropic and adrenergic control**	**Negative**	Heart failure	CPA [106]	BG9928 [107] Rolofylline [108] Capadenoson [122] Neladenoson [124]
	Dromotropic and chronotropic control	Negative	Arrhythmia	Tecadenoson (DC)[128] Selodenoson [129] PJ-875 [129]	CVT-2759 [130]
	Vascular control	Constriction	Hypertension		
	Ischemia	Protection	Angina		Capadenoson [121]
	Adaptation	Hypertrophy		CPA [106]	
		Angiogenesis			
			Diabetes mellitus and hyperlipidemia	GR79236 and ARA [165] SDZ WAG994 and RPR749 [123] Tecadenoson [168]	GS-9667/CVT-3619 [167] BW-1433 [168]
			Glaucoma	Trabodenoson (DC) [123]	
			Glomerulonephritis	Metrifudil 2 (DC) [123]	
**A_2A_AR**	Inotropic and adrenergic control	Positive	Heart failure	LASSBio-294 [184]	
	Vascular control	Dilatation	Imaging	Regadenoson, binodenoson, evodenoson, sonedenoson, and apadenoson [123,185]	
	Anti-inflammation		Platelet aggregation	PSB-15826, PSB-12404, and PSB-16301 [186]	
			Cholesterol homeostasis	ATL313 [187]	
	Ischemia	Protection	Angina	CGS 21680C [180] ATL-193/ATL-146e [181]	
	Adaptation	Hypertrophy			
		Angiogenesis			
**A_2B_AR**	Vascular control	Dilatation			
	Anti-inflammation		Atherosclerosis and hyperlipidemia	BAY 60-6853 [189]	
	Ischemia	Protection	Angina	BAY 60-6583 [190]	
	Adaptation	Hypertrophy			
		Angiogenesis			GS-6201 [191]
			Diabetes mellitus and hyperlipidemia		NECA [192] ATL-801 [193] MRS-1754 [194]
**A_3_AR**	Vascular control	Constriction			
	Anti-inflammation		Diabetic kidney diseases		LJ-2698 [195]
			Atherosclerosis and hyperlipidemia		LJ-1888 [196]
	Ischemia	Protection	Angina	IB-MECA [197] CI-IB-MECA [198] CP-532,903 [199]	
	Adaptation		Hypertrophy		
			Angiogenesis		

**Table 3 cells-09-00785-t003:** Adenosine receptor drugs in CNS diseases.

Receptors	Diseases	Drugs		
		Selective Full Agonist	Antagonist/Partial Agonist	Allosteric Modulators
**A_1_AR**	Stroke	NNC-21-0136 [123]		
	Sleep	CPA [155]		
	Anxiety and depression	MRS5474 [151]		TRR469 [150]
	Cognition and memory		ASP5854 [159]	
	Alzheimer’s disease			
	Huntington’s diseases	CPA, CCPA, CHA and ADAC [162]		
	Schizophrenia	2-CLA. NECA, CHA, CPA, PIA, L-PIA [210]	CPT, 8-CPT2, DPCPX, and MSX-3 [210]	
	Pain	GR79236X [138] Cl-ENBA [139] MRS7469 [140] SDZ WAG 994, GR79236, GW- 493838 (DC) [141]		T62 (DC) [141]
	Epilepsy		DPCPX [146]	
**A_2A_AR**	Stroke		SCH58261 [200]	
	Sleep	CGS21680 [133]		
	Anxiety and depression		Istradefylline [208]	
	Cognition and memory		ASP5854 [159]	
	Alzheimer’s disease		MSX-3 [212]	
	Huntington’s diseases		ZM241385 [220]	
	Parkinson’s diseases		Tozadenant/SYN115, DT1133, ZM241385, ST1535, and istradefylline [61] Preladenant, ASP5854, vipadenant (DC) [61] SCH900800, and BIIB014 [221] KW-6002 [134]	
	Schizophrenia	APEC, CGS21680, NECA, CV-1808, and DPMA [210]	MSX-3, DMPX, SCH58261, and ZM241385 [210]	
	Pain	CGS21680 [155] Spongosine (DC) [204]		
	Epilepsy		ZM241385 [202]	
	Drug addiction	CGS21680 and NECA [215]		
**A_3_AR**	Stroke	IB-MECA [222] CI-IB-MECA [223]	LJ-1251 [72]	
	Epilepsy		ANR235 [224]	

**Table 4 cells-09-00785-t004:** Adenosine receptor drugs in cancer.

Receptors	Diseases	Drugs
Selective Full Agonist	Antagonist/Partial Agonist
**A_2B_AR**	Bladder and breast cancer		ATL801 [251]
Melanoma		PSB1115 [252]
**A_3_AR**	Colon carcinoma	IB-MECA [320]	
Melanoma	IB-MECA [321]	
Prostate carcinoma	IB-MECA [322]	
HCC	CI-IB-MECA [323]

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
