# Peer review of "Focusing on Adenosine Receptors as a Potential Targeted Therapy in Human Diseases"

_cells, 2020, doi:10.3390/cells9030785_

Round 1

Reviewer 1 Report

This work is a collection of wide range effects mediated by adenosine through its receptors. Therapeutic effects of adeniosine receptor agonists and antagonists or moduators are described in the central nervous system and peripheral tissues.

I have found several issues in the text which need correction.

P2. line 93/94 - the sentence needs to be corrected - The conversion of AMP to adenosine is decreased because increase in its concentration inhibits this effect...

Adenosine metabolism is more complex mechanism - Authors should describe it more precisely (besides ADA, adenosine is catabolized vis adenosine kinase (AK) or incorporated to S-adenosylhomocysteine via SAHH)

P.3 line 118/120 - the sentence is not clear - PAM increases effect of endogenous ligand adenosine; make this clear

P.6 line 249 - ...all od ARs are involved in hypertrophy "of what"?

P. 8 line288/290 - how A1R hyperpolarization  is related neuroprotection; please expline it

P.8 line 306/307 - the sense of this statement is not clear

P8 line311/312 - the statement is not clear ...induced anxiety and aggression….?

P.8 line 326/330 - please, mention drugs evidenced in the table or refer table 3

P.13, chapter 4.2.4 - The effect on drug addiction is not mentioned in the text

P.13 line 474 - A2BAR might be active on high adenosine concentrations

In table 3 - add reference when mentioning agonists used to treat HD

P15 line 483 - endothelial cells

Im the whole text there are grammar errors; verbs are lacking, this often changes sense of the ststements

Author Response

Tatsuya Nagano, M.D., Ph.D.                   

Assistant Professor of                   

Division of Respiratory Medicine                           

Department of Internal Medicine                

Kobe University Graduate School of Medicine        

7-5-1 Kusunoki-cho, Chuo-ku,                  

Kobe City 650-0017, Japan                    

Phone: +81-78-382-5660                             

Fax: +81-78-382-5661                                 

E-mail: tnagano@med.kobe-u.ac.jp                           

March 18, 2020

Dear Dr Sybil Miao,

We are sending the revised version of our manuscript entitled "Focusing on Adenosine Receptors as a Potential Target Therapy in Human Diseases". We have addressed the critical issues raised by Reviewer and made appropriate changes to the revised version of our manuscript.

In addition, we have received English proofreading service and corrected the English in our manuscript.

We are confident that these modifications do not affect the conclusion of our manuscript. We really thank the Reviewer for constructive comments that help us to improve the quality of our presentation. We hope that the changes we made are enough to make this manuscript acceptable for publication in Cells.

Yours sincerely,

Tatsuya Nagano, M.D. Ph.D.

Reviewer 2 Report

The authors have undertaken a fairly comprehensive review of adenosine receptors for the development of pharmaceuticals in diseases.  I have some comments that I believe need to be addressed prior to publication of this article.  Especially, English grammar and syntax in the manuscripts should be checked and corrected by a native English-speaking person.

Major comment:

References 107 and 129 are duplicates.

References 214 and 222 are duplicates.

References 280 and 286 are duplicates.

References 275 and 288 are duplicates.

Minor comments:

Page 2 line 56, “S-adenosyl-homocysteinase”, is S-adenosyl-homocysteine.

Page 2 line 57, “S-adenosyl-homocysteinase”, is S-adenosyl-homocysteine.

Page 3 lines 114–115, “Receptors will interact with the endogenous adenosine agonist (orthosteric ligand) or with a molecule (allosteric ligand) that are topographically distinct from the orthosteric [49].”, is “the orthosteric site”.

Figure 1, “S-adenosyl-homocysteinase”, is S-adenosyl-homocysteine.

Figure 1, “S-adenosyl-homocysteinase hydrolase”, is S-adenosyl-homocysteine hydrolase.

Figure 1, “CD37”, is CD73.

Figure 1, “G1” close to A1AR, is Gi

Figure 1, “G1” close to A2AAR, is GS

Figure 1, “G1” close to A3AR, is Gi

Page 4 line 150, “S-adenosyl-homocysteinase through enzyme S-adenosyl-L-homocysteinase hydrolase.”, is S-adenosyl-homocysteine through enzyme S-adenosyl-L-homocysteine hydrolase.

Page 5 lines 203–206, “In addition, A1AR increasing NK activity, inducing O2- generation from eosinophil, releasing thromboxane A2 and IL-6 from endothelial, inducing chemotaxis dendritic cells (DC) and suppressing vesicular major histocompatibility complex (MHC) class I cross-presentation, and increasing endothelial permeability [23].”, Please revise this sentence.

Page 6 lines 207–209, “Pro-inflammatory effects A1AR in monocytes enhancing Fcγ receptor-mediated phagocytosis [78], secreting vascular endothelial growth factor (VEGF) [79], promoting monocyte differentiated into osteoclast [80].”, Please revise this sentence.

Page 6 lines 228–230, “Regardless of its discontinuing study, EPI-2010, an A1AR antisense oligonucleotide promoter region, has shown EPI-2010 to be safe and well-tolerated, with modest indications of efficacy in patients with mild asthma [89].”, Please revise this sentence.

Page 6 line 238, “G1”, is Gi.

Page 6 line 240, “HSP27”, Please define it.

Page 6 lines 247–248, “By interacting with A2AR, A1AR inhibited necrosis cardiac cell ischemia model [105] through phosphorylation of ERK1/2 [106].”, Please revise this sentence.

Page 6 line 252, “HF”, Please define it.

Page 6 line 252, “Rolofylline”, is rolofylline.

Table 1. “CCPA [92]” is agonist.

Table 1. “(DC)”, Please define it.

Page 7 line 263, “Capadenoson”, is capadenoson.

Page 7 line 264, “Neladenoson”, is neladenoson.

Page 7 line 265, “Neladenoson”, is neladenoson.

Page 7 lines 271–273, “Negative chronotropic and dromotropic involved inactivation of the inwardly rectifying K+ current (IK,Ado or IK,Ach), inhibition of the inward Ca2+ current (ICa), modification of If, activation of nitric oxide synthase (NOS) [107].”, Please revise this sentence.

Page 7 line 273, “Nevertheless, A1AR role either anti- or pro-arrhythmic [129].”, Please revise this sentence.

Page 7 line 277, “Tecadenoson”, is tecadenoson.

Page 7 line 280, “Selodenoson”, is selodenoson.

Page 8 line 285, “The concept of A1AR for neuroprotection still arguable.”, Please revise this sentence.

Page 8 lines 326–327, “Double activation of A1AR and A2AAR influenced glutamatergic and dopaminergic neurotransmission, the two neurotransmitter systems most affected by schizophrenia [163].”, Please revise this sentence.

Page 9 line 345, “Tecadenoson”, is tecadenoson.

Page 9 lines 345–346, “In addition, Tecadenoson may lowering FFA levels and relatively safe for the heart and an A1AR antagonist and also agonist A2BAR, BW-1433, improving glucose tolerance [171].”, Please revise this sentence.

Page 9 line 348, “Trabodenoson”, is trabodenoson.

Page 9 line 348, “Metrifudil”, is metrifudil.

Page 10 lines 383–386, “In asthmatic airways, A2AAR suppresses inflammation by reduced neutrophil adherence to the endothelium, inhibited fMLP-induced oxidative burst, inhibited superoxide anion generation, inhibited LPS-induced TNF-α expression [86], inhibited histamine and tryptase release, stimulated wound healing and VEGF secretion [87].”, Please revise this sentence.

Page 10 line 395, “Regadenoson”, is regadenoson.

Page 10 line 396, “Apedonosone”, is apedonosone.

Page 10 line 399, “Pentoxyfilline”, is pentoxyfilline.

Page 10 lines 410–411, “However, A2aAR not only giving protection against ischemic but also increased the occurrence of cardiac arrhythmias [185].”, Please revise this sentence.

Page 10 line 419, “Binodenoson”, is binodenoson.

Page 10 line 419, “Evodenoson”, is evodenoson.

Page 10 line 420, “Sonedenoson”, is sonedenoson.

Page 10 line 420, “Apadenoson”, is apadenoson.

Page 10 lines 424–426, “Furthermore, A2AAR be a center of study for the anti-inflammatory effects of adenosine against excess cholesterol accumulation in developing new cardiovascular therapies [191].”, Please revise this sentence.

Table 2. “(DC)”, Please define it.

Table 2. “LJ-2698 [198]” is antagonist.

Table 2. “LJ-1888 [199]” is antagonist.

Page 13 lines 432–433, “According to the physiological, neuromodulation of adenosine through activation of high-affinity A2AAR is important [165].”, Please revise this sentence.

Page 13 line 446, “Spongosine”, is spongosine.

Page 13 line 451, “Istradefylline”, is istradefylline.

Page 13 line 454, “A2AAR/D23”, is A2AAR/D3.

Page 13 line 466, “Tozadenant”, is tozadenant.

Page 13 line 467, “Istradefylline”, is istradefylline.

Table 3. “L-PIA [222]” is [214].

Table 3. “MSX-3 [222]” is [214].

Table 3. “(DC)”, Please define it.

Page 17 lines 550–553, “Following remodeling effect and vasodilatation, A2BAR may have a role in inhibiting smooth muscle cells-associated hypertension [245] and intriguingly, A2BAR KO mice had normal blood pressure [246]. Recently, the deletion of A2BAR protecting against salt-induced hypertension and stroke [247] and inhibiting an increase of mean arterial blood pressure [248].”, Please revise these sentences.

Page 17 lines 559–560, “Concerning DM, A2BAR modulating glucose homeostasis and lipid metabolism, insulin secretion and resistance, inflammation, β-cells survival and kidney protection [250].”, Please revise this sentence.

Page 17 line 561, “A2BAR”, is A2BAR agonist.

Page 17 lines 586–588, “Pro-inflammation role in mast cells exists by inducing the release of histamines and other allergic mediators [260], inhibiting of eosinophil chemotaxis in lung tissue [261], inhibiting apoptosis that involved the beta gamma subunits of Gi, PI3K/AKT [262].”, Please revise this sentence.

Page 18 lines 594–595, “In addition, A3AR up-regulated of lymphocytes T cell activation [269].”, Please revise this sentence.

Page 18 line 609, “It has a role in asthma, COPD, lung fibrosis, and pulmonary inflammation.”, Please define “it”.

Page 19 lines 649–650, “More, deletion A3AR protecting from the development of renal and cardiovascular disease [296].”, Please revise this sentence.

Page 19 line 658, “FC”, Please define it.

Page 19–20 lines 688–690, “A3AR also a potential candidate…”, Please revise this sentence.

Page 20 lines 714–715, “CI-IB-MECA able to increase retinal ganglion survival [323] and mediates a tonic proliferative effect in colon tumor cells [324].”, Please revise this sentence.

Page 25 Reference 82, Add page range: F369–F376.

Page 25 Reference 99, Page range: H1411–H1416.

Page 26 Reference 103, Page range: R693–R701.

Page 26 Reference 111, Page range: 183–188.

Page 26 Reference 113, Journal Name is Allergy

Page 27 Reference 125, Page range: 124.

Page 27 Reference 130, Add page range: H145–H153.

Page 27 Reference 131, Page range: 3202–3208.

Page 27 Reference 133, Year: 2015.

Page 27 Reference 138, Add page range: 676.

Page 28 Reference 149, Add page range: 235.

Page 28 Reference 151, Abbreviated Journal Name Year, Volume, page range: Eur. J. Neurosci. 2002, 16, 547–550

Page 29 Reference 177, Add page range: 146ra108.

Page 29 Reference 178, Add page range: 224.

Page 30 Reference 182, Add page range: H2364–H2372.

Page 30 Reference 184, Add page range: H3164–H3171.

Page 30 Reference 193, Add page range: 585297.

Page 30 Reference 198, Add page range: 38.

Page 31 Reference 200, Page range: H607–H613.

Page 31 Reference 206, Add page range: R31–R41.

Page 31 Reference 211, Add page range: S82–S87.

Page 31 Reference 213, Title of the article: Potential role of adenosine A2A receptors in the treatment of schizophrenia.

Page 31 Reference 216, Add page range: 235.

Page 32 Reference 224, Add page range: 658.

Page 32 Reference 227, Year: 2001.

Page 32 Reference 238, Add page range: 21.

Page 33 Reference 241, Add page range: H1183–H1189.

Page 33 Reference 242, Page range: 310.

Page 33 Reference 245, “(Dallas, Tex. 1979)”, should be deleted.

Page 33 Reference 245, Page range: 786–793.

Page 33 Reference 248, “FEBS J.”, is “FASEB J.”.

Page 33 Reference 253, Add page range: 37.

Page 33 Reference 256, “(United States)”, should be deleted.

Page 34 Reference 267, “(80-. ).”, should be deleted.

Page 34 Reference 273, Volume, page range: 2014, 708746

Page 34 Reference 274, Page range: R959–R969.

Page 34 Reference 280, Page range: 107–113. “e1137.” should be deleted.

Page 34 Reference 282, Add page range: 134.

Page 35 Reference 291, Page range: H1307–H1313.

Page 35 Reference 296, Add page range: e003868.

Page 35 Reference 297, Page range: H3448–H3455.

Page 36 Reference 314, Abbreviated Journal Name: Am. J. Physiol. Regul. Integr. Comp. Physiol.

Page 36 Reference 316, Abbreviated Journal Name: Am. J. Physiol. Renal. Physiol.

Page 37 Reference 330, Volume: 278

Page 37 Reference 331, Page range: 2077–2083.

Author Response

(The authors gave the same response as above.)

Round 2

Reviewer 1 Report

Authors added suggested detailes; the work is worth of publishing in MDPI journal.

Please, change the title in reference 210 to:

Involvement of adenosine A2A receptors in depression and anxiety

Author Response

Thank you for your comment. Please see the attachment.
